# MALDI-TOF MS-Based KPC Direct Detection from Patients’ Positive Blood Culture Bottles, Short-Term Cultures, and Colonies at the Hospital

**DOI:** 10.3390/pathogens12070865

**Published:** 2023-06-22

**Authors:** Agustina Costa, Roque Figueroa-Espinosa, Jerson A. Martínez, Liliana Fernández-Canigia, María I. Maldonado, Silvina A. Bergese, Ana E. Schneider, Carlos Vay, Carlos H. Rodriguez, Marcela Nastro, Gabriel O. Gutkind, José A. Di Conza

**Affiliations:** 1Instituto de Investigaciones en Bacteriología y Virología Molecular (IBaViM), Facultad de Farmacia y Bioquímica, Universidad de Buenos Aires, Buenos Aires 1113, Argentina; aguspcosta@gmail.com (A.C.); rfigues@gmail.com (R.F.-E.); andrwe14@gmail.com (J.A.M.); 2Consejo Nacional de Investigaciones Científicas y Técnicas (CONICET), Buenos Aires 1033, Argentina; 3Hospital Alemán de Buenos Aires, Buenos Aires 1113, Argentina; lfcanigia@hospitalaleman.com (L.F.-C.); ivanam27@gmail.com (M.I.M.); silvinabergese@gmail.com (S.A.B.); anaschneider89@gmail.com (A.E.S.); 4Hospital de Clínicas “José de San Martín”, Universidad de Buenos Aires, Buenos Aires 1118, Argentina; carlos.vay@smdei.com (C.V.); herrod3066@gmail.com (C.H.R.); marcelanastro@hotmail.com (M.N.)

**Keywords:** KPC, blood culture, short-term culture, MALDI-TOF MS

## Abstract

Carbapenemase resistance in *Enterobacterales* is a global public health problem and rapid and effective methods for detecting these resistance mechanisms are needed urgently. Our aim was to evaluate the performance of a MALDI-TOF MS-based “*Klebsiella pneumoniae* carbapenemase” (KPC) detection protocol from patients’ positive blood cultures, short-term cultures, and colonies in healthcare settings. Bacterial identification and KPC detection were achieved after protein extraction with organic solvents and target spot loading with suitable organic matrices. The confirmation of KPC production was performed using susceptibility tests and *bla*_KPC_ amplification using PCR and sequencing. The KPC direct detection (KPC peak at approximately 28.681 Da) from patients’ positive blood cultures, short-term cultures, and colonies, once bacterial identification was achieved, showed an overall sensibility and specificity of 100% (CI95: [95%, 100%] and CI95: [99%, 100%], respectively). The concordance between hospital routine bacterial identification protocol and identification using this new methodology from the same extract used for KPC detection was ≥92%. This study represents the pioneering effort to directly detect KPC using MALDI-TOF MS technology, conducted on patient-derived samples obtained from hospitals for validation purposes, in a multi-resistance global context that requires concrete actions to preserve the available therapeutic options and reduce the spread of antibiotic resistance markers.

## 1. Introduction

Carbapenem resistance in *Enterobacterales* (CRE) is a worldwide public health problem, whose magnitude was enlarged after the SARS-CoV-2 pandemic [1,2,3]. In Argentina, we also witnessed a significant increase in the rate of CRE during 2020, with an alarming emergence of multiple carbapenemases producers [4]. CRE represent a global epidemiological risk for healthcare systems and a serious threat to actual and future antimicrobial treatments [5,6,7]. *Klebsiella pneumoniae* carbapenemase (KPC), the most prevalent variants being KPC-2 and KPC-3, are by now the most commonly reported carbapenemase around the world and are associated with high morbidity and mortality rates. Their location on self-conjugative plasmids and frequent association with *K. pneumoniae* are some of the factors that contribute to their global dissemination [8].

Blood stream infection with CRE is associated with high mortality rates [9,10] and, as previously stated by Kumar et al. [11], the rapid instauration of adequate antibiotic therapy for bacteremia is crucial for patients’ prognoses, raising the need for new rapid methodologies for resistance detection to be developed.

Carbapenemase-producing *K. pneumoniae* is the most commonly isolated pathogen from rectal swabs when the surveillance of carbapenemase carriers in hospital closed units is carried out [12]. Screening is usually performed using chromogenic culture media, where carbapenem-resistant bacteria are recovered from a patient’s sample [13,14] and KPC confirmation is made afterwards using phenotypic synergy tests [15]. KPC producers are also commonly isolated from other types of clinical specimens, such as respiratory and urine samples [8,16].

KPC detection in clinical laboratories is typically accomplished using traditional phenotypic methods. Among these, synergy tests (using β-lactams and β-lactams inhibitors) via disk diffusion methods are commonly employed in low and moderate complexity facilities [15]. Additionally, colorimetric assays, such as a Blue Carba test [17] and Carba-NP [18], can be performed, but even though they are operator-friendly, they do not define the enzyme involved in the resistance mechanism and may not be attainable for every clinical laboratory. These culture-based methodologies are easy to perform but require the isolation of the pathogen on solid culture media after at least an 18–24 incubation period, and their sensibilities and specificities range from 84 to 100% and 91 to 100%, respectively [15]. Colorimetric assays to detect carbapenemase activity, along with bacterial identification, have also been tested using short-term cultures (STC) obtained from positive blood culture (BC) bottles showing good results [19,20], but no protocol has been evaluated directly from positive patients’ BC bottles. Additionally, lateral flow immunoassays are available for clinical laboratories, showing a high sensitivity and specificity, but most of them are generally expensive [15]. Carbapenemase genotypic detection (such as Polymerase Chain Reaction—PCR—assays or Whole Genome Sequencing—WGS) is highly sensitive but not commonly available in most clinical laboratories because of its elevated cost [21] and the necessity for trained personnel. Therefore, there is an ongoing effort to develop and validate new molecular and immunological methods for KPC detection in clinical settings.

Today, Matrix-assisted laser desorption ionization–time of flight mass spectrometry (MALDI-TOF MS) technology is extensively used worldwide for the microbial identification (ID) of bacteria and fungi [22,23]. Once the spectrometer is incorporated into the healthcare institution, the costs associated with sample processing are relatively low [23]. Antibiotic resistance detection is one of the current challenges that this technology faces [24]. Although MALDI-TOF MS hydrolysis assays have been described to detect carbapenemases, they are not commonly implemented in clinical laboratories [25]. The latest approaches in MALDI-TOF MS resistance detection involve the use of machine learning techniques to discriminate between resistant and susceptible isolates [26,27,28].

We formerly developed a methodology for CMY [29] and KPC [30] detection using MALDI-TOF MS from isolated colonies (COL), showing results with a high sensibility and specificity. The aim of this study was to evaluate the performance of a fast and easy bacterial identification and KPC detection protocol using MALDI-TOF MS from patients’ positive blood cultures, short-term cultures, and colonies in healthcare settings, testing its concordance with the results obtained in each hospital.

## 2. Materials and Methods

### 2.1. Control Strains

Recombinant strains (*E. coli* TOP10/pKPC-2 and *E. coli* TOP10/pKPC-3) [30,31] expressing the most prevalent KPC variants were used as controls, in order to establish the *m*/*z* value of the enzyme in the spectrum as a reference. Receptor strains (*E. coli* TOP10 and *E. coli* TOP10/pK19) not expressing the enzymes were evaluated as negative control spectra. Protein extraction with formic acid–isopropyl alcohol–water, 17:33:50 (*v*/*v*) (FA-ISO) was performed from isolated colonies (COL) on solid culture media [30]. *K. pneumoniae* ATCC 700603 was also used as a negative control strain.

### 2.2. KPC Detection from Simulated Positive Blood Cultures, Short-Term Cultures and Colonies Using Previously Characterized Isolates

We evaluated a panel of 93 *Enterobacterales* (60 *K. pneumoniae* and 33 *Escherichia coli*) for the bacterial identification and KPC detection from the simulated positive blood cultures and short-term cultures, and 118 *Enterobacterales* (60 *K. pneumoniae*, 28 *E. coli*, 12 *Enterobacter cloacae* complex, 3 *Citrobacter braakii*, and 15 *Serratia marcescens*) for the bacterial identification and KPC detection from the colonies (Table 1). 

All the isolates were previously characterized phenotypically using identification, disk diffusion tests, and sinergy tests [32], and genotypically using PCR and sequencing [33] at Laboratorio de Resistencia Bacteriana (Facultad de Farmacia y Bioquímica, Universidad de Buenos Aires).

The sample processing from the simulated positive BC and COL was performed as previously described by Figueroa-Espinosa et al. [30]. The sample processing from the STC was performed the same way as from the isolated colonies [30].

We evaluated the bacterial identification results and calculated the KPC detection sensibility and specificity from a visual inspection of the KPC-producing and non-KPC-producing isolates’ spectra for every type of sample.

### 2.3. Clinical Samples

A total of 193 samples, collected during a 7-month period between 2022 and 2023, were included in this study: 78 positive BC bottles, 78 STC samples, and 37 COL samples. The samples showing positive growth for members of *Enterobacterale*s were included for direct processing from the positive BC bottles (49 were analyzed at Hospital Alemán and 29 at Hospital de Clínicas). The bacterial identification and KPC detection were evaluated directly from the positive BC bottles (*n* = 78) and the corresponding STC (*n* = 78).

In addition, 37 carbapenem-resistant isolates recovered from rectal swabs (*n* = 25) and other clinical specimens (*n* = 12) were included. To test the bacterial identification and KPC detection from solid culture media, we selected blue colonies grown on CHROMagar^TM^ KPC supplemented with meropenem (CHROMagar, Paris, France) recovered from rectal swabs, and isolates obtained from urine cultures grown on Mueller Hinton Agar (Laboratorio Argentino, Buenos Aires, Argentina) showing resistance to carbapenems. All the samples from the colonies were analyzed at Hospital Alemán.

As the patients’ personal information was encrypted, this study was exempted from the requirement of written informed consent. Additionally, it was approved by the Ethics Committee of Facultad de Farmacia y Bioquímica (Universidad de Buenos Aires) (RESCD-2020-134-E-UBA-DCT_FFYB, 20 August 2020). 

### 2.4. Hospitals’ Bacterial Routine Identification

The bacterial routine identification at both hospitals was performed using the standard direct MALDI-TOF MS protocol [22] with STC obtained from positive BC bottles, and from isolated colonies for COL samples. A loopful of bacteria from the STC/colonies was laid onto a steel target plate with a wooden stick, then 1 µL of α-cyano-4-hydroxycinnamic acid (HCCA) was deposited on the spot. After drying at room temperature, an automatic analysis using MALDI-TOF MS’s flexAnalysis software was performed. 

### 2.5. Protein Extraction from Patients’ Positive Blood Culture Bottles

As described previously [30], the protein extraction was performed using organic solvents. Briefly, 1.4 mL of positive blood culture was transferred to an eppendorf tube, which was centrifuged at 1.4 rpm for 5 min. One milliliter of the supernatant was collected and centrifuged at 13,000 rpm for 2 min, then the pellet was washed once with 1 mL of distilled water, vortexed for 30 s, and centrifuged at 13,000 rpm for 2 min. The bacterial pellet was re-suspended in 300 mL of distilled water and vortexed for 30 s at room temperature. Then, 900 mL of absolute ethanol (Sigma-Aldrich, Burlington, MA, USA) was added, vortexed for 30 s, and centrifuged at 13,000 rpm for 2 min. The supernatant was discarded and the pellet was re-suspended in 100 µL of extraction solvent (FA-ISO) (Sigma-Aldrich, USA). The suspension was vortexed for 30 s and centrifuged for 2 min at 13,000 rpm. The supernatant extract was used both for the bacterial identification and KPC detection using MALDI-TOF MS.

### 2.6. Protein Extraction from Short-Term Cultures

In addition to processing the samples directly from the BC bottles, protein extraction was also performed for the corresponding STC. Two drops (approximately 100 µL) of positive BC were plated onto Blood Agar plates and incubated at 37 °C for 4–5 h, in a 5% CO_2_ atmosphere. Protein extracts from STC were obtained using the FA-ISO extraction method, as previously described by Figueroa-Espinosa et al. [30], on isolated colonies, but after a shorter incubation period (4–5 h instead of 18–24 h). The supernatant extract was used both for the bacterial identification and KPC detection using MALDI-TOF MS. 

### 2.7. Protein Extraction from Colonies

The protein extraction was performed according to the Figueroa-Espinosa et al. [30] protocol (FA-ISO extraction method) from the isolated colonies on CHROMagar^TM^ KPC, which was supplemented with meropenem and lawns grown on Mueller Hinton Agar after 18–24 h of incubation at 37 °C. Supernatant extracts were used both for the bacterial identification and KPC detection using MALDI-TOF MS.

### 2.8. Target Spot Loading for Bacterial Identification and KPC Detection 

For the bacterial identification, 1 µL of protein extract was co-crystallized with 1 µL of HCCA matrix and analyzed after being dried at room temperature (one spot per sample). 

For the KPC detection, protein extracts obtained from patients’ positive BC, STC, and COL were spotted onto the steel target plate using a double-layer sinapinic acid (SA) method, as follows: first, a layer of 0.7 µL of an SA-saturated solution (10 mg/mL SA in absolute ethanol) (Sigma-Aldrich, USA) was laid on the spot; after drying at room temperature, a second layer of 1 µL of an SA solution in acetonitrile (30:70 *v*/*v*) (Sigma-Aldrich, USA) and 0.1% trifluoroacetic acid (Sigma-Aldrich, USA) in water was deposited above the first one, and finally, 1 µL of the protein extract was added in the final step. The samples were left to dry at room temperature and then analyzed using MALDI-TOF MS. For the KPC detection, each extract was analyzed in duplicate. 

Additionally, ferulic acid (FA) matrix [34,35] was evaluated for the KPC detection: 1 µL of the protein extract was laid on the spot and 1 µL of an FA solution (12.5 mg/mL in acetonitrile–formic acid–distilled water 33:17:50) was added afterwards. Each extract was analyzed in duplicate. 

### 2.9. Spectra Acquisition

For the bacterial identification in the low molecular weight range, spectra were obtained in the linear positive ion mode of a Microflex LT mass spectrometer (Bruker Daltonics, Bremen, Germany) with flexControl 3.4 software (Bruker Daltonics, Germany), using the automatic MBT_FC.par method with default parameters. Before each run, the spectrometer was calibrated using Bacterial Test Standard (Bruker Daltonics, Germany).

For the high molecular weight range analysis (KPC detection), spectra were obtained in the linear positive ion mode of the Microflex LT mass spectrometer (Bruker Daltonics, Germany) with flexControl 3.4 software, using the LP44_44kDa.par method.

At Hospital Alemán, the parameters were configured as follows: mass range: from 10,000 Da to 50,000 Da; spectrometer ion source 1: 19.99 kV; ion source 2: 17.94 kV; lens: 5.99 kV; pulsed ion extraction: 650 ns; detection gain: 3017 V; laser frequency: 60 Hz; and laser power: 90%. Each spectrum was obtained after 1000–1200 shots per spot.

At Hospital de Clínicas, the parameters were configured as follows: mass range: from 10,000 Da to 50,000 Da; spectrometer ion source 1: 19.94 kV; ion source 2: 17.78 kV; lens: 5.95 kV; pulsed ion extraction: 650 ns; detection gain: 2745 V; laser frequency: 60 Hz; and laser power: 90%. Each spectrum was obtained after 1000–1200 shots per spot.

The data were manually acquired using autoXecute mode at both hospitals. Before each run, the spectrometer was calibrated using Protein Standard II Calibration Mix (Bruker Daltonics, Germany), containing a mixture of Protein A and Trypsinogen.

### 2.10. Bacterial Identification Concordance Calculation

The bacterial identification concordance between the hospital routine method (direct MALDI-TOF MS protocol from STC or isolated colonies [22]) and the protein extraction using the FA-ISO method was calculated according to the following formula [36]:Concordance = (No. of result matches/total tests) × 100

When different species belonged to the *E. cloacae* complex, the group as a whole (the complex) was considered for the concordance analysis, regardless of species name, as recommended by the Argentinian National Network for Microbiological Identification by Mass Spectrometry [37].

### 2.11. Visual Spectra Analysis and Statistics for KPC Detection

The spectra obtained directly from patients’ positive BC, STC, and COL were analyzed visually using the flexAnalysis 3.4 software (Bruker Daltonics, Germany). We searched for the visual presence/absence of KPC peaks in every spectrum after baseline subtraction and smoothing, considering the expected size of the enzyme observed when analyzing the control strains as reference. In addition, we evaluated the intensity in the *y* axis (arbitrary units) for every spectrum in the expected KPC *m*/*z* position.

We also visually searched for a ~11,109 Da peak, which had previously been reported to be associated with a common KPC dissemination platform [38,39,40] on spectra acquired for bacterial identification with HCCA.

Only those samples for which bacterial identification with the FA-ISO extraction method was achieved were included for the KPC detection statistical analysis with ClinPro Tools, as we considered that failed identification was indicative of a low efficiency in the protein extraction process, which could lead to false negative results when detecting KPC for a producing culture.

The spectra of each protein extract were analyzed after automatic calibration and normalization with the software ClinPro Tools 3.0 (Bruker Daltonics) [41]. The statistical analysis was performed using the full raw spectra (10,000 to 50,000 Da) of the duplicates with the “Peak Statistic Calculation” tool. The area under the curve (AUC) of the ROC curve was evaluated for the selected peak between the KPC-producing and non-producing strains to determine the discriminative power, and the Genetic Algorithm (GA) was used to calculate the sensitivity and specificity.

### 2.12. Antimicrobial Susceptibility Testing and Genetic Characterization of Isolates

All the isolates were characterized phenotypically using disk diffusion tests according to the Clinical and Laboratory Standards Institute (CLSI) guidelines [42] at the hospitals. The production of carbapenemases was investigated with synergy tests using boronic acid (BOR), EDTA, and carbapenems [32] at both hospitals. A Blue Carba Test (bioMérieux, Craponne, France) [17] was performed at Hospital de Clínicas on most STC obtained from the positive BC bottles when requested by physicians, and an immunochromatographic assay (Buenos Aires, Argentina) [15] for carbapenemase detection was performed for two isolates at Hospital Alemán. The results obtained from the hospitals remained blinded and were not shared until the conclusion of the study, ensuring a double-blind approach.

Genotypic characterization was carried out at Laboratorio de Resistencia Bacteriana (Facultad de Farmacia y Bioquímica, Universidad de Buenos Aires) using PCR amplification performed on the total DNA using the primers and conditions described previously [33]. We searched for the carbapenemase-encoding genes usually found in *Enterobacterales* (*bla*_KPC_ and *bla*_NDM_), and *bla*_KPC_ amplicons were sequenced on both strands using an ABI3730XL DNA Sequencer (Macrogen, Seoul, Republic of Korea).

## 3. Results

### 3.1. KPC-Producing Recombinant Strains Spectra Analysis

The KPC *m*/*z* observed in the control strains’ spectra obtained using SA as matrix was 28,679 Da for KPC-2 (Figure 1a) and 28,703 Da for KPC-3 (Figure 1b), with both peaks being absent for the receptor strains. These were considered as a reference for the visual evaluation of the clinical samples. The KPC variants’ *m*/*z* values were similar when analyzed at both hospitals. 

### 3.2. Bacterial Identification and KPC Detection from Simulated Positive Blood Cultures, Short-Term Cultures and Colonies Evaluated with Previously Characterized Isolates

The bacterial identification performed using the FA-ISO extraction method showed a complete concordance with the previous characterization results for all the isolates evaluated (93 *Enterobacterales* from simulated positive BC and STC and 118 from colonies).

For the visual detection of the KPC (peak ~28,680 Da) from the simulated positive BC and STC, both the sensitivity and specificity were 100% (CI95%: [90%–100%] for sensibility, CI95%: [93%–100%] for specificity).

Regarding the KPC visual detection from the isolated colonies, the sensitivity and specificity were also 100% (CI95%: [93%–100%] for sensibility; CI95%: [94%–100%] for specificity).

### 3.3. Bacterial Identification from Clinical Samples

Bacterial identification from the patients’ positive BC bottles using the FA-ISO extraction method was achieved in 71/78 samples. Seven BC samples rendered a not reliable identification (NRI) result, probably due to a low efficiency in the protein extraction process. 

Considering the samples for which a successful identification was achieved from the patients’ BC bottles (*n* = 71), 41 samples were identified as *K. pneumoniae* by the hospital protocol, whereas the identification results using the FA-ISO extraction method were *K. pneumoniae* (*n* = 39) and *K. variicola* (*n* = 2); 1 sample identified as *K. variicola* by the hospital was identified as *K. pneumoniae* by the FA-ISO protocol; *E. coli* was concordantly identified in every case (*n* = 24), as well as *S. marcescens* (*n* = 5), the *E. cloacae* complex (*n* = 1), and *Proteus mirabilis* (*n* = 1) (Table 2). The concordance rate for bacterial identification directly from patients’ positive BC bottles reached 98% when considering the samples for which a successful identification was achieved.

Regarding the bacterial identification from the STC (*n* = 78), 45 samples identified by the hospital as *K. pneumoniae* were identified as *K. pneumoniae* (*n* = 40) and *K. variicola* (*n* = 5) by the FA-ISO extraction method. *E. coli* (*n* = 24), the *E. cloacae* complex (*n* = 2), *K. variicola* (*n* =1), and *P. mirabilis* (*n* = 1) were concordantly identified by both methods. Five samples were identified as *S. marcescens* by the hospital, whereas the FA-ISO method identification result showed *S. marcescens* (*n* = 4) and *S. ureilytica* (*n* = 1) (Table 3). The concordance for the bacterial identification from the STC was 92%.

When comparing the bacterial identification from the COL samples (n = 37) between the direct MALDI-TOF MS method on the isolated colonies (hospital routine identification) and the FA-ISO extraction method, there was a 100% concordance. Specifically, 36 isolates were identified as *K. pneumoniae* and one of them was identified as *Proteus mirabilis* (Table 4).

### 3.4. KPC Detection from Patients’ Positive BC Bottles

The KPC producers (7 *K. pneumoniae*) evaluated directly from the patients’ positive BC bottles showed a peak between 28,655 Da and 28,740 Da (median = 28,722, CI95: [28,655 Da; 28,740 Da]) (Figure 2a). This peak was absent in the non-KPC producers’ spectra (*n* = 64), although some intensity in the KPC *m*/*z* range was observed in some cases (background noise), but did not constitute a clear peak (Table 2). The intensities ranged from 221 a.u. to 1095 a.u. for the KPC producers (median = 610 a.u.) and from 0 a.u. to 176 a.u. (median = 27.5 a.u.) for the non-KPC producing strains (Figure 3a).

For the calculation of the statistical parameters, we considered only the samples for which bacterial identification was achieved using the FA-ISO extraction method, indicating an efficient protein extraction process (see Section 2.11). 

The statistical analysis with the ClinPro Tools showed a significant difference (*p*-value < 0.001) between the KPC producers and non-KPC producers for a selected peak at 28,724 Da. The AUC of the ROC curve for this specific peak was 0.98, indicating a great discrimination power between the groups. The sensibility and specificity for the KPC detection from the positive BC bottles, calculated using the GA tool, was 100% for both parameters (CI95%: [77%; 100%] for sensibility; CI95%: [97%; 100%] for specificity) (Table 5).

### 3.5. KPC Detection from STC Samples

The KPC producers (8 *K. pneumoniae*) evaluated from the STC obtained from the positive BC bottles showed a peak between 28,660 Da and 28,728 Da (median = 28,676 Da, CI95%: [28,660 Da–28,728 Da]) (Figure 2b). This peak was absent in the non-KPC producers’ spectra (*n* = 70), although some intensity in the KPC *m*/*z* range was observed in some of them, but this did not constitute a clear peak (Table 3). The intensities ranged from 261 a.u. to 1283 a.u. for the KPC producers (median = 835.5 a.u.) and from 0 a.u. to 520 a.u. (median = 48.5 a.u.) for the non-KPC-producing strains (Figure 3b).

The statistical analysis with the ClinPro Tools showed a significant difference (*p*-value < 0.001) between the KPC producers and non-KPC producers for a selected peak at 28,679 Da. The AUC of the ROC curve for this specific peak was 0.97, indicating a great discrimination power between the groups. The sensibility and specificity for the KPC detection from the STC, calculated using the GA tool, was 100% for both parameters (CI95%: [79%; 100%] for sensibility; CI95%: [97%; 100%] for specificity) (Table 5).

### 3.6. KPC Detection from COL Samples

All the *K. pneumoniae* KPC producers evaluated from the COL (*n* = 18) showed a peak between 28,662 and 28,716 Da (median = 28,683 Da, CI95%: [28,676 Da–28,698 Da]) (Figure 2c), and the intensities ranged from 727 to 3949 a.u. (median = 1503.5 a.u.) (Figure 3c). This peak was not present in the non-KPC producers’ spectra (*n* = 19). When evaluating the intensities in the KPC *m*/*z* range for the COL for non-KPC producers, values in the range of 4–519 a.u. were observed (median = 67 a.u.) (Table 4).

The statistical analysis with the ClinPro Tools showed a significant difference (*p*-value < 0.001) between the KPC producers and non-KPC producers for a selected peak at 28,686 Da. The AUC of the ROC curve for this specific peak was 0.97, indicating a great discrimination power between the groups. The sensibility and specificity for the KPC detection from the COL, calculated using the GA tool, was 100% for both parameters (CI95%: [90% 100%] for sensibility; CI95%: [91%–100%] for specificity) (Table 5).

A comparison of the KPC *m*/*z* median, intensity median for the samples containing KPC producers and non-KPC producers, and sensibility and specificity for each type of sample evaluated in this study is shown in Table 5. The median *m*/*z* value for the KPC in all the KPC-producing samples (7 BC, 8 STC, and 18 COL) was 28,681 Da (CI95%: [28,676 Da–28,687 Da]), with a median intensity of 1095 a.u. In contrast, the median intensity for the non-KPC producers was only 45 a.u. The overall sensibility and specificity for the KPC detection were 100%, [CI95%: 95%; 100% and CI95%: 99%; 100%, respectively].

### 3.7. Peak at m/z ~11,109 Da Visual Detection 

Considering all the samples included in this study (BC, STC, and COL), a ~11,109 Da peak, corresponding to the P019 protein associated with Tn*4401a* transposon carried by some KPC-producing *K. pneumoniae* strains, was visually detected on 21/33 KPC producers’ spectra, ranging from 11,100 to 11,117 Da (median: 11,109 Da, [CI95%: 11,106 Da; 11,110 Da]), and it was not detected in the non-KPC producing isolates (Table 2, Table 3 and Table 4). This biomarker was successfully detected in the KPC producers from 3/7 positive BC bottles, 3/8 STC samples, and 15/18 COL samples, and its overall sensibility and specificity were 63% and 100%, respectively.

### 3.8. Ferulic Acid Matrix Performance for KPC Detection

The spectra acquired from all the protein extracts (BC bottles, STC, and COL) after loading the FA target spots showed similar results to those obtained with the SA. Moreover, the KPC peak detection was successfully achieved when FA was used as a co-crystallization matrix. Although background noise was observed when analyzing some non-KPC-producing isolates spectra, no distinct peaks in the KPC *m*/*z* range were observed when the FA matrix was employed (Figure 4).

### 3.9. Antimicrobial Susceptibility Testing and Carbapenemase Gene Detection by PCR

Twenty-six *K. pneumoniae* STC (18 *K. pneumoniae*, 5 *S. marcescens*, 2 *E. coli*, and 1 *E. cloacae* complex) were tested via Blue Carba tests at Hospital de Clínicas, and 12 of them rendered a positive result for carbapenemase production. When analyzed using PCR amplification, four of them were *bla*_KPC_ carriers and eight were *bla*_NDM_ carriers. Two samples (1HC15 and 1HC16) rendered a negative Blue Carba test result and *bla*_NDM_ was amplified using a PCR afterwards. The two isolates tested using immunochromatography at Hospital Alemán (HC64 and HC65) rendered a *bla*_KPC_ positive result via PCR. 

All the KPC-producing isolates (8 BC/STC samples and 18 COL samples) showed cephalosporin and carbapenem resistance with disk diffusion tests. In addition, positive amplification for *bla*_KPC_ was obtained via a PCR assay. In total, 34 samples (10 BC/10 STC samples and 24 COL samples) showed positive amplification only for *bla*_NDM_ and 7 isolates (COL) were co-carriers of *bla*_KPC_ and *bla*_NDM_ (Table 2, Table 3 and Table 4). Two clinical isolates were positive for *bla*_IMP_ amplification (Table 4). Regarding the KPC variants, 21 isolates (8 BC/8 STC samples and 13 COL samples) carried *bla*_KPC-2_ and 5 isolates (COL samples) carried *bla*_KPC-3_ (Table 2, Table 3 and Table 4). Non-carbapenem resistant isolates rendered a negative result for the carbapenemase genes amplification, as expected.

## 4. Discussion

Previous studies have evaluated KPC detection from isolated colonies and simulated positive blood cultures using MALDI-TOF MS [30,43,44]. This is the first study including patients’ samples that was performed in the healthcare setting. Bacterial identification and KPC detection were successfully achieved from liquid culture media (BC broth), as well as from different solid culture media (Blood Agar, chromogenic media and Mueller Hinton Agar). 

Moreira et al. reported a sensibility of 98.09% and specificity of 97.9% for KPC detection from isolated colonies [43] and a sensibility of 94.9% and specificity of 95.3% for KPC detection from simulated positive blood cultures [44]. However, it is worth noting that these statistical parameters were slightly lower than the ones reported in this study when evaluating the previously characterized isolates from our strain collection.

Regarding the patients’ samples, we proved that KPC detection from BC bottles and STC can be achieved during the first hours after the BC bottle becomes positive, reducing the turnaround time (TAT) of 24–48 h for traditional KPC phenotypic verification methods (used at clinical laboratories) (Figure 5). We strongly recommend conducting KPC detection solely after achieving a successful bacterial identification from the protein extract in the first place. This parameter acts as a “check-point” to ensure the efficiency of the protein extraction process before proceeding with the KPC detection. If the bacterial identification is unsuccessful, we suggest repeating the protein extraction protocol if the sample is still available. 

MALDI-TOF MS KPC peak detection from rectal swab isolates, as well as isolates recovered from other clinical specimens, can significantly reduce the TAT of commonly used phenotypic synergy tests. This accelerated approach can assist in the clinical decision making process of isolating patients with KPC fecal carriage in hospital closed units (Figure 5).

For the KPC-2 producers, a KPC peak at approximately 28,681 Da was consistently detected after the visual analysis of the spectra of the KPC-producing samples, compared to the reference *m*/*z* observed in the control strains (*E. coli* TOP10/pKPC-2), with the exceptions of 1HD19, HC63, HC64, HD65, and POR15 (KPC-2 carriers). For these samples, the observed *m*/*z* values in the spectra were higher than those in the control strain spectrum. Given the potential variation in the *m*/*z* value of the KPC peak, it is advisable to adopt an *m*/*z* range for the detection instead of a fixed peak value. The slight difference between the manually calculated KPC peak median values and the KPC peak values selected with the ClinPro Tools program may be attributed to the software parameters. 

Previous studies have reported different KPC *m*/*z* values. Yoon et al. [45] estimated a KPC *m*/*z* of 28.718 Da when analyzing transformant strains after protein extraction with a lysis buffer from colonies grown in MacConkey Agar. Regarding the spectrometer parameters, this research group used a pulse ion extraction of 1200 ns. Moreira et al. [43] evaluated different pulse ion extraction settings using the same FA-ISO extraction method evaluated in this study [30], finding different KPC *m*/*z* ranges in each scenario. Performing various protein extraction methodologies and different acquisition parameters could then lead to different KPC *m*/*z* values. Nonetheless, Moreira et al. [43,44] reported a similar KPC peak range when analyzing isolated colonies and artificial positive BC, which aligns with the range observed for the samples analyzed in this study.

Although a direct comparison between both KPC detection approaches (mature protein versus biomarker) was not performed, it is evident that the KPC confirmation using KPC peak detection exhibited a higher sensitivity when compared to the biomarker approach (which relied on the detection of the previously reported ~11,109 Da peak). High sensitivity and specificity values for the KPC prediction have been reported for this peak in the United States [38,46] and Europe [39]. We previously observed a low sensitivity for KPC prediction using this biomarker in our region [30], which might be attributed to a different scenario for the circulation of the genetic platform responsible for its presence (Tn*4401*a). Undoubtedly, the KPC peak detection strategy shows a much higher sensitivity than the ~11,109 Da peak approach, at least for the set of samples evaluated in this study.

Additionally, we observed that, while some samples contained non-KPC-producing bacteria and exhibited spectra with background noise at the KPC *m*/*z* position, a clear visual distinction could still be made between these spectra and the spectra with a clear KPC-peak. Although the KPC *m*/*z* intensity could be used as a parameter for differentiating the KPC producers from the non-KPC producers [30,43], we believe that a visual inspection of the spectra is still necessary, as automated software may not be able to distinguish between background noise and a true KPC-peak.

As previously mentioned by Moreira et al. [44], it is important to highlight that the equipment parameters should be tested and optimized in every MALDI-TOF spectrometer for KPC detection before analyzing patients’ samples. For this purpose, it would be ideal to evaluate control strains beforehand. In this study, we demonstrated a successful detection of the KPC peak using two different spectrometers in two different hospitals. To obtain reproducible results, the training of clinical laboratory staff would be relevant. We also recommend performing replicates of the target spot loading, as the results may vary between spots, due to possible differential protein co-crystallization. 

We highlight the capacity of this protocol both to detect KPC presence and to identify the etiological agent from the same protein extract obtained using FA-ISO from BC, STC, and COL. Protein extracts can be used for bacterial identification and detecting the ~11,109 Da peak by analyzing the low molecular weight range, as well as detecting the KPC peak in the high molecular weight range, selecting the appropriate organic matrices and acquisition parameters. Additionally, this protocol can be implemented with commonly used chemical reagents and simple centrifugation and separation steps, making it an easy to perform methodology. It is also important to acknowledge that this methodology has its inherent limitations. Firstly, it was not feasible to use spectra intensities to discriminate between the KPC-producing and non-producing isolates. Secondly, the assay required an optimization of the equipment parameters and the use of a different calibrator and organic matrix for the protein detection compared to those employed for the bacterial identification. Lastly, the current methodology has a limited capacity for detecting a single enzyme (KPC) at this stage.

Ferulic acid matrix, which has previously been reported for high molecular weight protein detection [34,35], could be used as an alternative organic matrix for detecting the KPC peak in the high molecular weight range, as its performance was similar to that of sinapinic acid, giving more options to clinical laboratories in terms of available chemical reagents.

Clinical laboratories can incorporate this new rapid and simple methodology for KPC detection in selected samples on a daily basis, based on local epidemiology. The method is easily implementable and can be used to test any bacterial culture suspected of carbapenemase production, thus expanding the already established utility of MALDI-TOF MS. Unfortunately, the current methodology is unable to detect the NDM enzyme, probably due to its inefficiency in extracting membrane-anchored proteins [47]. However, in the future, it would be valuable to explore hybrid approaches that integrate machine learning techniques [26,27,28] with our KPC peak detection method to categorize CRE isolates based on the specific type of resistance mechanism they exhibit.

## 5. Conclusions

MALDI-TOF MS technology has significant potential in clinical settings for detecting antibiotic resistance, particularly in a multi-resistance global context. Once the spectrometer is acquired by the hospital or institution, this fast and effective MALDI-TOF MS protocol can quickly confirm KPC production, reducing the turnaround time compared to traditional phenotypic methods, optimizing the use of the available antibiotics and improving patients’ prognoses. Healthcare settings could benefit from a faster detection of resistance markers, thereby preserving the available therapeutic options and reducing the spread of antibiotic resistance. In conclusion, this study represents the pioneering effort to directly detect KPC using MALDI-TOF MS technology, conducted on patient-derived samples obtained from hospitals for validation purposes.

## Figures and Tables

**Figure 1 pathogens-12-00865-f001:**
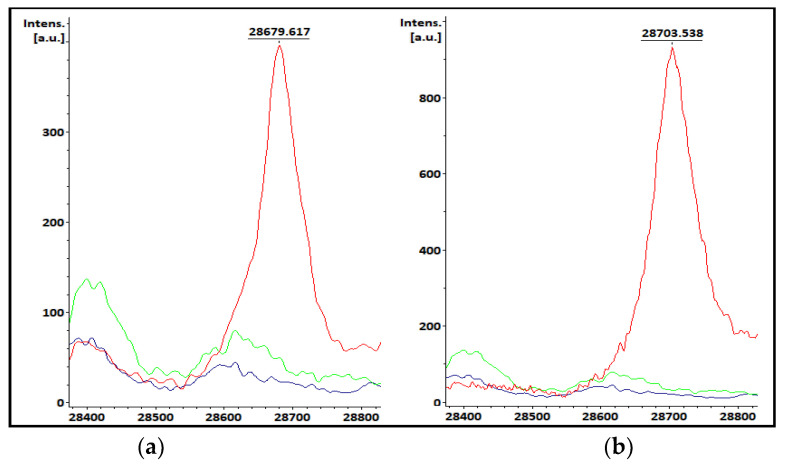
Control strains spectra. Recombinant strains spectra expressing (**a**) KPC-2 or (**b**) KPC-3 are shown in red and receptor strains spectra are shown in blue (*E. coli* TOP10) and green (*E. coli* TOP10+pK19).

**Figure 2 pathogens-12-00865-f002:**
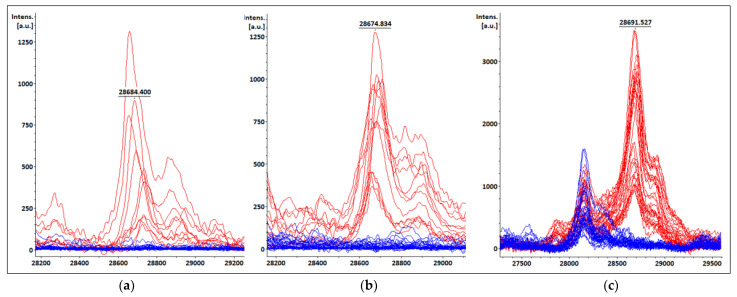
Spectra obtained from (**a**) patients’ positive BC bottles; (**b**) STC; and (**c**) COL. KPC peaks on KPC producers’ spectra are shown in red and spectra from samples containing non-KPC-producing bacteria are shown in blue. The KPC *m*/*z* value of one spectrum is displayed as an example.

**Figure 3 pathogens-12-00865-f003:**
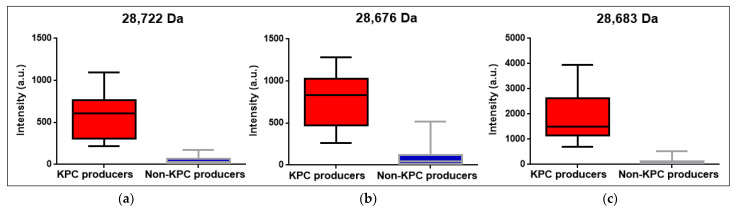
Box plots showing median and interquartile range 95% for spectra intensities at KPC *m*/*z* obtained from (**a**) BC, (**b**) STC, and (**c**) COL. Intensities for KPC *m*/*z* of spectra obtained from samples containing KPC producers and non-KPC producers are shown in red and blue boxes, respectively. KPC *m*/*z* shown value corresponds to the median calculated for every type of sample.

**Figure 4 pathogens-12-00865-f004:**
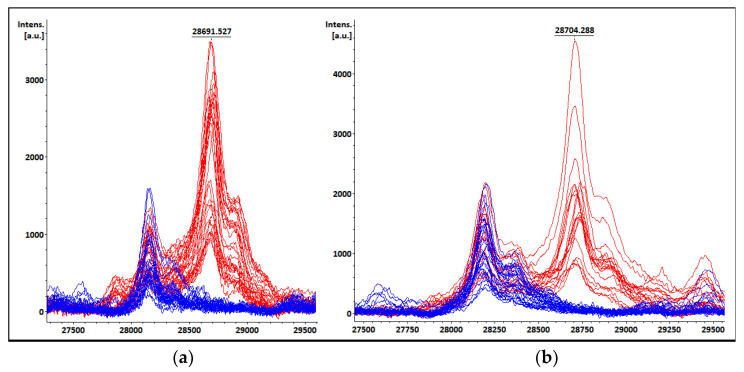
Comparison of spectra after target spot loading with (**a**) SA and (**b**) FA. KPC-producers’ spectra are shown in red (red) and non-KPC producers’ spectra are shown in blue. The KPC *m*/*z* value of one spectrum is displayed as an example.

**Figure 5 pathogens-12-00865-f005:**
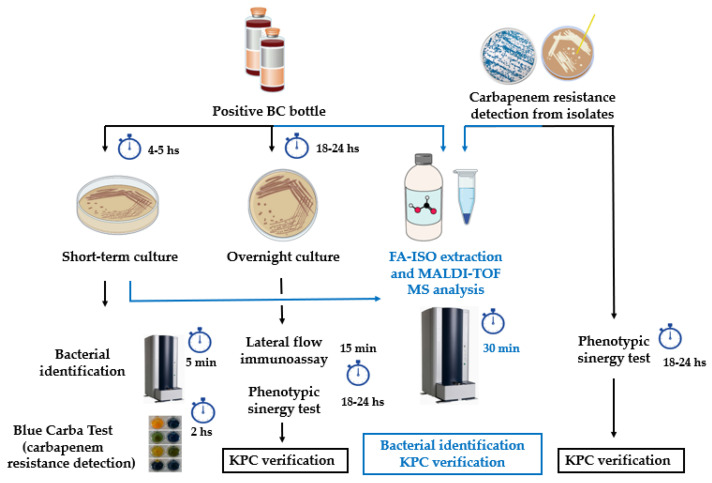
Comparison of turnaround time for phenotypic KPC confirmation methodologies (black lines) and MALDI-TOF MS detection with FA-ISO extraction method (blue lines) from positive blood culture bottles, short-term cultures, and colonies.

**Table 1 pathogens-12-00865-t001:** Bacterial isolates evaluated from simulated positive blood cultures, short-term cultures, and isolated colonies.

Species	Total Isolates	KPC-Producing Isolates	Non-KPC-Producing Isolates
Simulated positive blood cultures and short-term cultures			
*K. pneumoniae*	60	32	28
*E. coli*	33	5	28
Total	93	37	56
Isolated colonies			
*K. pneumoniae*	60	39	21
*E. coli*	28	4	24
*E. cloacae* complex	12	7	5
*C. braakii*	3	1	2
*S. marcescens*	15	4	11
Total	118	55	63

**Table 2 pathogens-12-00865-t002:** Bacterial identification, KPC peak *m*/*z* and intensity of spectra after visual analysis from patients’ positive BC bottles. KPC mass value and intensities correspond to the average *m*/*z* for both spectra duplicates. Spectra intensities for KPC non-producing isolates were calculated considering the position of KPC peak *m*/*z* for KPC-producing isolates median.

Sample	Hospital Routine ID	ID from BC	KPC Peak *m*/*z* from BC (Da)	Intensity (a.u.)	Peak at *m*/*z* ~11,109 Da	Resistance Markers
1HD02	*K. pneumoniae*	*K. pneumoniae*	28,687	762	(+)	*bla* _KPC-2_
1HD19	*K. pneumoniae*	*K. pneumoniae*	28,722	221	Absent	*bla* _KPC-2_
1HD21	*K. pneumoniae*	*K. pneumoniae*	28,655	1095	(+)	*bla* _KPC-2_
1HD22	*K. pneumoniae*	*K. pneumoniae*	28,660	613	(+)	*bla* _KPC-2_
HD63	*K. pneumoniae*	*K. pneumoniae*	28,736	355	Absent	*bla* _KPC-2_
HD64	*K. pneumoniae*	*K. pneumoniae*	28,725	311	Absent	*bla* _KPC-2_
HD65	*K. pneumoniae*	*K. pneumoniae*	28,740	610	Absent	*bla* _KPC-2_
1HD03 *	*K. pneumoniae*	NRI	NA	NA	NA	*bla* _KPC-2_
HD01	*K. pneumoniae*	*K. pneumoniae*	Absent	27	Absent	NCD
HD02	*K. pneumoniae*	*K. pneumoniae*	Absent	130	Absent	NCD
HD03	*K. pneumoniae*	*K. pneumoniae*	Absent	50	Absent	NCD
HD04	*K. pneumoniae*	*K. pneumoniae*	Absent	40	Absent	NCD
HD06	*K. pneumoniae*	*K. pneumoniae*	Absent	11	Absent	NCD
HD07	*K. pneumoniae*	*K. pneumoniae*	Absent	0	Absent	NCD
HD08	*E. coli*	*E. coli*	Absent	0	Absent	NCD
HD10	*E. coli*	*E. coli*	Absent	30	Absent	NCD
HD11	*E. coli*	*E. coli*	Absent	33	Absent	NCD
HD12	*E. coli*	*E. coli*	Absent	28	Absent	NCD
HD14	*E. coli*	*E. coli*	Absent	80	Absent	NCD
HD15	*K. pneumoniae*	*K. pneumoniae*	Absent	154	Absent	NCD
HD16	*E. coli*	*E. coli*	Absent	24	Absent	NCD
HD18	*E. coli*	*E. coli*	Absent	143	Absent	NCD
HD20*	*E. coli*	NRI	NA	NA	NA	NCD
HD21	*E. coli*	*E. coli*	Absent	62	Absent	NCD
HD22	*E. coli*	*E. coli*	Absent	69	Absent	NCD
HD23	*K. pneumoniae*	*K. variicola*	Absent	0	Absent	NCD
HD24	*K. pneumoniae*	*K. pneumoniae*	Absent	20	Absent	NCD
HD25*	*K. pneumoniae*	NRI	NA	NA	NA	NCD
HD26*	*E. coli*	NRI	NA	NA	NA	NCD
HD27*	*K. pneumoniae*	NRI	NA	NA	NA	NCD
HD28	*E. coli*	*E. coli*	Absent	34	Absent	NCD
HD29	*E. coli*	*E. coli*	Absent	78	Absent	NCD
HD30	*E. coli*	*E. coli*	Absent	8	Absent	NCD
HD33	*K. pneumoniae*	*K. variicola*	Absent	106	Absent	NCD
HD35	*P. mirabilis*	*P. mirabilis*	Absent	88	Absent	NCD
HD36	*E. coli*	*E. coli*	Absent	125	Absent	NCD
HD40	*K. pneumoniae*	*K. pneumoniae*	Absent	65	Absent	NCD
HD42	*E. coli*	*E. coli*	Absent	33	Absent	NCD
HD43	*E. coli*	*E. coli*	Absent	122	Absent	NCD
HD44	*E. coli*	*E. coli*	Absent	68	Absent	NCD
HD45	*E. coli*	*E. coli*	Absent	6	Absent	NCD
HD47	*K. pneumoniae*	*K. pneumoniae*	Absent	71	Absent	NCD
HD48	*K. pneumoniae*	*K. pneumoniae*	Absent	55	Absent	NCD
HD49	*K. pneumoniae*	*K. pneumoniae*	Absent	69	Absent	NCD
HD50	*K. pneumoniae*	*K. pneumoniae*	Absent	29	Absent	NCD
HD51	*K. pneumoniae*	*K. pneumoniae*	Absent	87	Absent	NCD
HD52*	*K. pneumoniae*	NRI	NA	NA	NA	NCD
HD53	*E. coli*	*E. coli*	Absent	29	Absent	NCD
HD54	*K. variicola*	*K. pneumoniae*	Absent	15	Absent	NCD
HD56	*K. pneumoniae*	*K. pneumoniae*	Absent	97	Absent	NCD
HD57	*K. pneumoniae*	*K. pneumoniae*	Absent	64	Absent	NCD
HD59	*E. cloacae* complex	*E. hormaechei*	Absent	176	Absent	NCD
HD61	*E. coli*	*E. coli*	Absent	21	Absent	NCD
HD62	*E. coli*	*E. coli*	Absent	24	Absent	NCD
1HD01 *	*E. cloacae* complex	NRI	NA	NA	NA	NCD
1HD04	*S. marcescens*	*S. marcescens*	Absent	7	Absent	NCD
1HD05	*S. marcescens*	*S. marcescens*	Absent	20	Absent	NCD
1HD06	*K. pneumoniae*	*K. pneumoniae*	Absent	2	Absent	NCD
1HD07	*K. pneumoniae*	*K. pneumoniae*	Absent	7	Absent	NCD
1HD08	*K. pneumoniae*	*K. pneumoniae*	Absent	7	Absent	*bla* _NDM_
1HD09	*E. coli*	*E. coli*	Absent	7	Absent	NCD
1HD10	*E. coli*	*E. coli*	Absent	8	Absent	NCD
1HD11	*K. pneumoniae*	*K. pneumoniae*	Absent	7	Absent	*bla* _NDM_
1HD12	*K. pneumoniae*	*K. pneumoniae*	Absent	5	Absent	*bla* _NDM_
1HD13	*K. pneumoniae*	*K. pneumoniae*	Absent	6	Absent	*bla* _NDM_
1HD14	*K. pneumoniae*	*K. pneumoniae*	Absent	5	Absent	*bla* _NDM_
1HD15	*K. pneumoniae*	*K. pneumoniae*	Absent	47	Absent	*bla* _NDM_
1HD16	*K. pneumoniae*	*K. pneumoniae*	Absent	55	Absent	*bla* _NDM_
1HD17	*S. marcescens*	*S. marcescens*	Absent	11	Absent	NCD
1HD18	*S. marcescens*	*S. marcescens*	Absent	46	Absent	NCD
1HD20	*K. pneumoniae*	*K. pneumoniae*	Absent	14	Absent	NCD
1HD23	*K. pneumoniae*	*K. pneumoniae*	Absent	9	Absent	NCD
1HD24	*K. pneumoniae*	*K. pneumoniae*	Absent	12	Absent	*bla* _NDM_
1HD25	*K. pneumoniae*	*K. pneumoniae*	Absent	4	Absent	*bla* _NDM_
1HD26	*K. pneumoniae*	*K. pneumoniae*	Absent	12	Absent	*bla* _NDM_
1HD27	*S. marcescens*	*S. marcescens*	Absent	11	Absent	NCD
1HD28	*K. pneumoniae*	*K. pneumoniae*	Absent	6	Absent	NCD
1HD29	*K. pneumoniae*	*K. pneumoniae*	Absent	9	Absent	NCD

* Samples not included in statistical analysis for KPC peak detection due to failed identification result, indicating a low efficiency in protein extraction process. Samples beginning with “1HD” and “HD” were analyzed at Hospital de Clínicas and Hospital Alemán, respectively. ID: bacterial identification, BC: blood culture, NRI: not reliable identification (score < 1.4), a.u.: arbitrary units, NA: not applicable, NCD: no carbapenemase genes detected (*bla*_KPC_/*bla*_NDM_).

**Table 3 pathogens-12-00865-t003:** Bacterial identification, KPC peak *m*/*z* and intensity of spectra after visual analysis from STC. KPC mass value and intensities correspond to the average *m*/*z* for both spectra duplicates. Spectra intensities for KPC non-producing isolates were calculated considering the position of KPC peak *m*/*z* for KPC-producing isolates median.

Sample	Hospital Routine ID	ID from STC	KPC Peak *m*/*z* from STC (Da)	Intensity (a.u.)	Peak at *m*/*z* ~11,109 Da	Resistance Markers
1HC02	*K. pneumoniae*	*K. pneumoniae*	28,686	1031	(+)	*bla* _KPC-2_
1HC03	*K. pneumoniae*	*K. pneumoniae*	28,679	1013	(+)	*bla* _KPC-2_
1HC19	*K. pneumoniae*	*K. pneumoniae*	28,728	1283	Absent	*bla* _KPC-2_
1HC21	*K. pneumoniae*	*K. pneumoniae*	28,675	823	(+)	*bla* _KPC-2_
1HC22	*K. pneumoniae*	*K. pneumoniae*	28,675	848	Absent	*bla* _KPC-2_
HC63	*K. pneumoniae*	*K. pneumoniae*	28,660	637	Absent	*bla* _KPC-2_
HC64	*K. pneumoniae*	*K. pneumoniae*	28,662	415	Absent	*bla* _KPC-2_
HC65	*K. pneumoniae*	*K. pneumoniae*	28,676	261	Absent	*bla* _KPC-2_
HC01	*K. pneumoniae*	*K. pneumoniae*	Absent	49	Absent	NCD
HC02	*K. pneumoniae*	*K. pneumoniae*	Absent	102	Absent	NCD
HC03	*K. pneumoniae*	*K. pneumoniae*	Absent	117	Absent	NCD
HC04	*K. pneumoniae*	*K. pneumoniae*	Absent	70	Absent	NCD
HC06	*K. pneumoniae*	*K. pneumoniae*	Absent	427	Absent	NCD
HC07	*K. pneumoniae*	*K. pneumoniae*	Absent	18	Absent	NCD
HC08	*E. coli*	*E. coli*	Absent	149	Absent	NCD
HC10	*E. coli*	*E. coli*	Absent	24	Absent	NCD
HC11	*E. coli*	*E. coli*	Absent	103	Absent	NCD
HC12	*E. coli*	*E. coli*	Absent	50	Absent	NCD
HC14	*E. coli*	*E. coli*	Absent	45	Absent	NCD
HC15	*K. pneumoniae*	*K. pneumoniae*	Absent	121	Absent	NCD
HC16	*E. coli*	*E. coli*	Absent	47	Absent	NCD
HC18	*E. coli*	*E. coli*	Absent	37	Absent	NCD
HC20	*E. coli*	*E. coli*	Absent	23	Absent	NCD
HC21	*E. coli*	*E. coli*	Absent	124	Absent	NCD
HC22	*E. coli*	*E. coli*	Absent	39	Absent	NCD
HC23	*K. pneumoniae*	*K. variicola*	Absent	405	Absent	NCD
HC24	*K. pneumoniae*	*K. variicola*	Absent	445	Absent	NCD
HC25	*K. pneumoniae*	*K. variicola*	Absent	192	Absent	NCD
HC26	*E. coli*	*E. coli*	Absent	48	Absent	NCD
HC27	*K. pneumoniae*	*K. variicola*	Absent	63	Absent	NCD
HC28	*E. coli*	*E. coli*	Absent	45	Absent	NCD
HC29	*E. coli*	*E. coli*	Absent	334	Absent	NCD
HC30	*E. coli*	*E. coli*	Absent	112	Absent	NCD
HC33	*K. pneumoniae*	*K. variicola*	Absent	22	Absent	NCD
HC35	*P. mirabilis*	*P. mirabilis*	Absent	158	Absent	NCD
HC36	*E. coli*	*E. coli*	Absent	311	Absent	NCD
HC40	*K. pneumoniae*	*K. pneumoniae*	Absent	329	Absent	NCD
HC42	*E. coli*	*E. coli*	Absent	163	Absent	NCD
HC43	*E. coli*	*E. coli*	Absent	116	Absent	NCD
HC44	*E. coli*	*E. coli*	Absent	106	Absent	NCD
HC45	*E. coli*	*E. coli*	Absent	330	Absent	NCD
HC47	*K. pneumoniae*	*K. pneumoniae*	Absent	52	Absent	NCD
HC48	*K. pneumoniae*	*K. pneumoniae*	Absent	157	Absent	NCD
HC49	*K. pneumoniae*	*K. pneumoniae*	Absent	60	Absent	NCD
HC50	*K. pneumoniae*	*K. pneumoniae*	Absent	103	Absent	NCD
HC51	*K. pneumoniae*	*K. pneumoniae*	Absent	93	Absent	NCD
HC52	*K. pneumoniae*	*K. pneumoniae*	Absent	15	Absent	NCD
HC53	*E. coli*	*E. coli*	Absent	3	Absent	NCD
HC54	*K. variicola*	*K. variicola*	Absent	10	Absent	NCD
HC56	*K. pneumoniae*	*K. pneumoniae*	Absent	111	Absent	NCD
HC57	*K. pneumoniae*	*K. pneumoniae*	Absent	265	Absent	NCD
HC59	*E. cloacae* complex	*E hormaechei*	Absent	520	Absent	NCD
HC61	*E. coli*	*E. coli*	Absent	13	Absent	NCD
HC62	*E. coli*	*E. coli*	Absent	30	Absent	NCD
1HC01	*E. cloacae* complex	*E. cloacae*	Absent	13	Absent	NCD
1HC04	*S. marcescens*	*S. marcescens*	Absent	4	Absent	NCD
1HC05	*S. marcescens*	*S. ureilytica*	Absent	11	Absent	NCD
1HC06	*K. pneumoniae*	*K. pneumoniae*	Absent	2	Absent	NCD
1HC07	*K. pneumoniae*	*K. pneumoniae*	Absent	3	Absent	NCD
1HC08	*K. pneumoniae*	*K. pneumoniae*	Absent	5	Absent	*bla* _NDM_
1HC09	*E. coli*	*E. coli*	Absent	16	Absent	NCD
1HC10	*E. coli*	*E. coli*	Absent	9	Absent	NCD
1HC11	*K. pneumoniae*	*K. pneumoniae*	Absent	7	Absent	*bla* _NDM_
1HC12	*K. pneumoniae*	*K. pneumoniae*	Absent	13	Absent	*bla* _NDM_
1HC13	*K. pneumoniae*	*K. pneumoniae*	Absent	0	Absent	*bla* _NDM_
1HC14	*K. pneumoniae*	*K. pneumoniae*	Absent	13	Absent	*bla* _NDM_
1HC15	*K. pneumoniae*	*K. pneumoniae*	Absent	94	Absent	*bla* _NDM_
1HC16	*K. pneumoniae*	*K. pneumoniae*	Absent	62	Absent	*bla* _NDM_
1HC17	*S. marcescens*	*S. marcescens*	Absent	38	Absent	NCD
1HC18	*S. marcescens*	*S. marcescens*	Absent	9	Absent	NCD
1HC20	*K. pneumoniae*	*K. pneumoniae*	Absent	34	Absent	NCD
1HC23	*K. pneumoniae*	*K. pneumoniae*	Absent	25	Absent	NCD
1HC24	*K. pneumoniae*	*K. pneumoniae*	Absent	46	Absent	*bla* _NDM_
1HC25	*K. pneumoniae*	*K. pneumoniae*	Absent	91	Absent	*bla* _NDM_
1HC26	*K. pneumoniae*	*K. pneumoniae*	Absent	83	Absent	*bla* _NDM_
1HC27	*S. marcescens*	*S. marcescens*	Absent	31	Absent	NCD
1HC28	*K. pneumoniae*	*K. pneumoniae*	Absent	9	Absent	NCD
1HC29	*K. pneumoniae*	*K. pneumoniae*	Absent	16	Absent	NCD

Samples beginning with “1HC” and “HC” were analyzed at Hospital de Clínicas and Hospital Alemán, respectively. ID: bacterial identification, STC: short-term culture, a.u.: arbitrary units, NCD: no carbapenemase genes detected (*bla*_KPC_/*bla*_NDM_).

**Table 4 pathogens-12-00865-t004:** Bacterial identification, KPC peak *m*/*z* and intensity of spectra after visual analysis from COL. KPC mass value and intensities correspond to the average *m*/*z* for both spectra duplicates. Spectra intensities for KPC non-producing isolates were calculated considering the position of KPC peak *m*/*z* for KPC-producing isolates median.

Sample	Hospital Routine ID	ID from COL	KPC Peak *m*/*z* from COL (Da)	Intensity (a.u.)	Peak at *m*/*z* ~11,109 Da	Resistance Markers
POR02	*K. pneumoniae*	*K. pneumoniae*	28,662	1734	(+)	*bla*_KPC-2_, *bla*_NDM_
POR03	*K. pneumoniae*	*K. pneumoniae*	28,685	1203	(+)	*bla*_KPC-2_, *bla*_NDM_
POR06	*K. pneumoniae*	*K. pneumoniae*	28,712	735	(+)	*bla* _KPC-3_
POR08	*K. pneumoniae*	*K. pneumoniae*	28,679	1158	(+)	*bla* _KPC-2_
POR09	*K. pneumoniae*	*K. pneumoniae*	28,710	1227	(+)	*bla* _KPC-3_
POR11	*K. pneumoniae*	*K. pneumoniae*	28,686	3184	(+)	*bla*_KPC-2_, *bla*_NDM_
POR14	*K. pneumoniae*	*K. pneumoniae*	28,716	2567	(+)	*bla* _KPC-3_
POR15	*K. pneumoniae*	*K. pneumoniae*	28,704	2829	(+)	*bla*_KPC-2_, *bla*_NDM_
POR16	*K. pneumoniae*	*K. pneumoniae*	28,676	3949	(+)	*bla*_KPC-2_, *bla*_NDM_
POR17	*K. pneumoniae*	*K. pneumoniae*	28,681	2386	(+)	*bla*_KPC-2_, *bla*_NDM_
POR18	*K. pneumoniae*	*K. pneumoniae*	28,687	2555	(+)	*bla* _KPC-3_
POR21	*K. pneumoniae*	*K. pneumoniae*	28,706	2963	(+)	*bla* _KPC-3_
POR23	*K. pneumoniae*	*K. pneumoniae*	28,685	1086	(+)	*bla* _KPC-2_
POR29	*K. pneumoniae*	*K. pneumoniae*	28,675	727	Absent	*bla*_KPC-2_, *bla*_NDM_
UC16	*K. pneumoniae*	*K. pneumoniae*	28,668	1321	(+)	*bla* _KPC-2_
UC24	*K. pneumoniae*	*K. pneumoniae*	28,677	1095	Absent	*bla* _KPC-2_
UC75	*K. pneumoniae*	*K. pneumoniae*	28,675	1546	(+)	*bla* _KPC-2_
COL65	*K. pneumoniae*	*K. pneumoniae*	28,680	1461	Absent	*bla* _KPC-2_
POR01	*K. pneumoniae*	*K. pneumoniae*	Absent	120	Absent	*bla* _NDM_
POR04	*K. pneumoniae*	*K. pneumoniae*	Absent	211	Absent	*bla* _NDM_
POR07	*K. pneumoniae*	*K. pneumoniae*	Absent	4	Absent	*bla* _NDM_
POR10	*K. pneumoniae*	*K. pneumoniae*	Absent	29	Absent	*bla* _NDM_
POR19	*K. pneumoniae*	*K. pneumoniae*	Absent	519	Absent	*bla* _NDM_
POR20	*K. pneumoniae*	*K. pneumoniae*	Absent	72	Absent	*bla* _NDM_
POR22	*K. pneumoniae*	*K. pneumoniae*	Absent	403	Absent	*bla* _NDM_
POR26	*K. pneumoniae*	*K. pneumoniae*	Absent	17	Absent	*bla* _NDM_
POR27	*K. pneumoniae*	*K. pneumoniae*	Absent	105	Absent	*bla* _NDM_
POR28	*K. pneumoniae*	*K. pneumoniae*	Absent	74	Absent	*bla* _NDM_
POR30	*K. pneumoniae*	*K. pneumoniae*	Absent	27	Absent	*bla* _NDM_
UC20	*K. pneumoniae*	*K. pneumoniae*	Absent	65	Absent	*bla* _NDM_
UC40	*K. pneumoniae*	*K. pneumoniae*	Absent	58	Absent	*bla* _IMP_
UC42	*K. pneumoniae*	*K. pneumoniae*	Absent	60	Absent	*bla* _NDM_
UC47	*K. pneumoniae*	*K. pneumoniae*	Absent	140	Absent	*bla* _NDM_
UC76	*K. pneumoniae*	*K. pneumoniae*	Absent	20	Absent	*bla* _NDM_
UC104	*K. pneumoniae*	*K. pneumoniae*	Absent	67	Absent	*bla* _IMP_
UC105	*K. pneumoniae*	*K. pneumoniae*	Absent	73	Absent	*bla* _NDM_
UC107	*P. mirabilis*	*P. mirabilis*	Absent	18	Absent	*bla* _NDM_

All COL samples were analyzed at Hospital Alemán. ID: bacterial identification, COL: isolated colonies, a.u.: arbitrary units.

**Table 5 pathogens-12-00865-t005:** Overall statistical results and comparison of calculated parameters for each type of sample (BC, STC, and COL).

Parameter	Positive BC	STC	COL	Overall
KPC *m*/*z* median	28,722 Da CI95%: [28,655 Da; 28,740 Da]	28,676 Da CI95%: [28,660 Da; 28,728 Da]	28,683 Da CI95%: [28,676 Da; 28,698 Da]	28,681 Da CI95%: [28,676 Da; 28,687 Da]
KPC *m*/*z* intensity median and range (KPC producers)	610 a.u. (221 a.u.–1095 a.u.)	835.5 a.u. (261 a.u.–1283 a.u.)	1503.5 a.u. (727 a.u.–3949 a.u.)	1095 a.u. (221 a.u.–3949 a.u.)
KPC *m*/*z* intensity median and range (non-KPC producers)	27.5 a.u. (0 a.u.–176 a.u.)	48.5 a.u. (0 a.u.–520 a.u.)	67 a.u. (4 a.u.–519 a.u.)	46 a.u. (0 a.u.–520 a.u.)
KPC detection sensibility	100% CI95%: [77%; 100%]	100% CI95%: [79%; 100%]	100% CI95%: [90%; 100%]	100% CI95%: [95%; 100%]
KPC detection specificity	100% CI95%: [97%; 100%]	100% CI95%: [97%; 100%]	100% CI95%: [91%; 100%]	100% CI95%: [99%; 100%]

BC: blood culture, STC: short-term culture, COL: isolated colonies, a.u.: arbitrary units.

## Data Availability

The data presented in this study are available on request to the corresponding author.

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
