# Peer review of "MALDI-TOF MS-Based KPC Direct Detection from Patients’ Positive Blood Culture Bottles, Short-Term Cultures, and Colonies at the Hospital"

_pathogens, 2023, doi:10.3390/pathogens12070865_

Round 1

Reviewer 1 Report

Overall the approach is interesting and should be implemented in clinical settings. However, some reworking is needed on the overall presentation. The introduction section should be made more clear.

Line 18: Abbreviations should be opened up when used for the first time: KPC

Lines 34-35: Authors should elaborate on how this magnitude enlargement has something to do with SARS-CoV-2, and not processes that have started already before the SARS-CoV-2. What is the proportion due to SARS-CoV-2 and what is due to previously set processes that are just continuing to get worse? The claim in these sentences seems to be detached from the rest of the paper, since SARS-CoV-2 is only mentioned once.

Line 37: Why is carbapenemase interesting other than having high morbidity and mortality rates?

Lines 41-42: there seems to be some verb missing, the sentence is hard to understand.

Lines 53-56: aren't MALDI-TOF MS) also extremely expensive?

Quality should be improved. Especially check the sentence structures to make the text easier to read.

Reviewer 2 Report

The research provides a promising alternative way to detect and report antibiotic-resistant isolates of Klebsiella pneumoniae. MALDI-TOF Mass Spectrometry provided the promising results when it was tested to detect the simulated samples and clinical samples.
The write-up of the manuscript is sound.

Topic is original and adds to the required knowledge. Recently, MALDI-TOF Mass Spectrometry has been commercially used to provide rapid diagnosis of the causative agents of disease. However, the current study extrapolated the objectives to report resistant isolates to declare alarming pathogens. The results obtained are much faster as compared to the conventional culturing, biochemical tests and molecular identification.

The methodology is sufficient. Simulated samples are used as positive controls as compared to clinical samples.  Conclusions are justified.
Tables and figures are providing the key findings of the study. However, there is need for improvement on the following:

Title can be improved by modifying to MALDI-TOF MS based KPC ....

Discussion can be improved by mentioning limitations of the study.

Few citations are suggested from 2023 year as well.

Its good

Round 2

Reviewer 1 Report

Good corrections to the manuscript.